



# Reduction in vehicular emissions attributable to the Covid-19 lockdown in Shanghai: insights from 5-year monitoring-based machine learning

Meng Wang[1], Yusen Duan[2], Zhuozhi Zhang[1], Qi Yuan[1], Xinwei Li[1], Shuwen Han[1], Juntao Huo[2], Jia Chen[2], Yanfen Lin[2], Qingyan Fu[2, *], Tao Wang[1], Junji Cao[3,4], Shun-cheng Lee[1, *]

[1]Department of Civil and Environmental Engineering, The Hong Kong Polytechnic University, Hung Hom, Hong Kong SAR, China
[2]Shanghai Environmental Monitoring Center, Shanghai, China
[3]State Key Laboratory of Loess and Quaternary Geology, Institute of Earth Environment, Chinese Academy of Sciences, Xi'an 710061, China
[4]Key Laboratory of Middle Atmosphere and Global Environment Observation, Institute of Atmospheric Physics, Chinese Academy of Sciences, Beijing 100029, China

*Correspondence to*: shun-cheng.lee@polyu.edu.hk (S.C. Lee) and qingyanf@sheemc.cn (Q.Y. Fu).

**Abstract.** Exposure to element carbon (EC) and $NO_x$ is a public health issue that has been gaining increasing interest, with high exposure levels generally observed in traffic environments e.g., roadsides. Shanghai, home to approximately 25 million in the Yangtze River Delta (YRD) region in east China, has one of the most intensive traffic activities in the world. However, our understanding of the trend in vehicular emissions and, in particular, in response to the strict Covid-19 lockdown is limited partly due to a lack of long-term observation dataset and application of advanced mathematical models. In this study, $NO_x$ and EC were continuously monitored at a near highway sampling site in west Shanghai for 5 years (2016-2020). The long-term dataset was used to train the machine learning model, rebuilding the $NO_x$ and EC in a business-as-usual (BAU) scenario in 2020. The reduction in $NO_x$ and EC attributable to lockdown was found to be smaller than it appeared because the first week of lockdown overlapped with the lunar new year holiday, whereas, at a later stage of lockdown, the reduction (50-70%) attributable to the lockdown was more significant, confirmed by satellite monitoring of $NO_2$. In contrast, the impact of the lockdown on vehicular emissions cannot be well represented by simply comparing the concentration before and during the lockdown for conventional campaigns. This study demonstrates the value of continuous air pollutant monitoring at a roadside on a long-term basis. Combined with the advanced mathematical model, air quality changes upon future emission control and/or event-driven scenarios are expected to be better predicted.

## 1 Introduction

As a response to the Covid-19 outbreak, strict lockdown measures were initiated in major cities across China in 2020, including the megacity of Shanghai in the Yangtze River Delta (YRD) region (He et al., 2020; Wang et al., 2020; Dai et al., 2021; Wu et al., 2021). The lockdown measures generally started in late January and lasted roughly one month, during which normal human activities were constrained substantially (He et al., 2020; Wang et al., 2020). The lockdown measures, such as shutting down cross-city travel and requiring people to stay at home, were strictly implemented to minimize human activities





(Zhao et al., 2020; Liu et al., 2020). As a result of these restrictive measures, anthropogenic emissions
of air pollutants, in particular, vehicular emissions, have been found to been reduced substantially as
evidenced by the evolution of $NO_2$ which is routinely measured at the ground air quality monitoring site,
as well as from the satellite monitoring (He et al., 2020; Li et al., 2021; Wu et al., 2021).
The impacts of vehicular emissions of $NO_2$ on public health are significant both through direct harm
on inhalation and as a precursor to secondary pollutants such as ozone and particulate matter (PM) (Lin
et al., 2022b; Lyu et al., 2022; Li et al., 2019; Lu et al., 2019). Although $NO_2$ concentrations are regulated
by air quality standards, limitations of $NO_x$ ($NO+NO_2$) emissions are becoming new emission standards
for new vehicles (Grange et al., 2017). In addition to $NO_x$ emission, on-road vehicles were also the major
source of primary PM emission, comprising various organic and inorganic species (Lin et al., 2018;
Hallquist et al., 2009; Fuzzi et al., 2015; Lin et al., 2020). As a major component of fine PM with a
diameter of less than 2.5 μm ($PM_{2.5}$), elemental carbon (EC) or black carbon is emitted a result of
incomplete combustion of fossil fuel (gasoline and diesel) in the internal combustion engine (Lin et al.,
2020; Lin et al., 2022a; Jia et al., 2021), with significant health and climate implications (Ramanathan
and Carmichael, 2008; Cappa et al., 2012; Rappazzo et al., 2015). With the recent implementation of
high emission standards (e.g., China IV and V), gasoline vehicles are generally less polluted, in terms of
EC emission when compared to diesel vehicles (Lin et al., 2020; Huang et al., 2022), especially with the
recent implementation of high emission standards (e.g., China IV and V). Gasoline-powered vehicles are
currently comprising over 90% of the total vehicles in China, with the trend of phasing out of vehicles
with old emission standards (i.e., China I–III) (Wang et al., 2019; Wang et al., 2022a). Nevertheless, on-
road vehicular emissions are still one of the major sources of $NO_x$ and EC in urban China (Zheng et al.,
2018; Zhang et al., 2019). Moreover, the total vehicular emission is also impacted by traffic mix and
volume, vehicle ages, and vehicle speed, while meteorological variables e.g., wind speed and wind
direction can impact the measured concentrations of air pollutants, making the quantification of vehicular
emission challenging in the real-world ambient environment.
The strict Covid-19 lockdown measures provided a unique opportunity to study the changes in event-
driven vehicular emissions, formulating a scientific basis for designing future air quality mitigation
strategies. However, the degree of reduction in vehicular emissions that can be attributable to the Covid-
19 outbreak varied greatly in different studies (up to over two-fold differences; (Jia et al., 2020; Wang
et al., 2020; Dai et al., 2021; Wu et al., 2021)). For example, by directly comparing the NOx
concentrations before and during the Covid-19 lockdown period, Jia et al. (2020) found a 56-58%
reduction in $NO_x$ during the Covid-19 lockdown period in Shanghai. However, the lockdown period
overlapped with the Chinese Spring Festival holiday (Wang et al., 2020), during which human activities
including traffic were already largely reduced. Moreover, meteorological conditions (e.g., wind speed
and direction) may vary, and, therefore, the direct comparison between two different periods does not
necessarily reflect the trend in emissions. To decouple the meteorological effects, a meteorological
normalization or de-weathering process was first proposed by Grange and Carslaw (2019) using a tree-
based machine learning algorithm. Vu et al. (2019) developed the de-weathering process to investigate
the seasonal trend of typical air pollutants routinely measured in Beijing and the de-weathered pollutants
showed a good agreement with the primary emission from the emission inventory. Using a similar de-





weathering process and taking into account the holiday effects. Dai et al. (2021) showed that the
reduction (-15.4%) in $NO_2$ attributable to Covid-19 lockdown was, on average, roughly half of the total
reduction (-29.5%) from comparing the measured and counterfactual $NO_2$ in a business as usual (BAU)
scenario during the overlapping period in 31 major Chinese cities. The decline in $NO_2$ attributable to the
lockdowns was also shown to be not as large as expected in 11 cities globally after a de-weathering
process (Shi et al., 2021). However, most of these tree-based machine learning studies did not quantify
the importance of the input variables, making these the machine learning process non-explainable or like
a "black box" (Lin et al., 2022b; Wang et al., 2022a) An explainable machine learning algorithm such as
the SHapley Additive exPlanation (SHAP) can quantify the impact of meteorological variables
(Lundberg et al., 2020; Qin et al., 2022; Wang et al., 2022a). However, few studies have applied the
explainable machine learning algorithm to study the trend in vehicular emissions. Moreover, most
previous studies focused on the changes in the measured $NO_2$ concentrations, which was routinely
measured in air quality monitoring site (He et al., 2020; Wang et al., 2020), while few studies reported
vehicular EC emissions based on long-term (years) measurement, and therefore, limiting our
understanding of vehicular $PM_{2.5}$ emissions under such a policy intervention and more importantly our
ability to predict future air quality changes upon similar emission control strategies.

95       Shanghai is an economic center of China, acting as a major transport hub. In 2019, the number of
civilian vehicles was over 4 million in Shanghai, approximately 13% higher than that in 2017 (Ministry
of Transport, 2020). On average, the daily ridership in Shanghai was over 57 million, with the turnover
quantity of motor vehicles of approximately 235 million passenger car unit kilometers (Ministry of
Transport, 2020). Because of the intensive traffic activities, exposure to EC has become a public health
issue that has been gaining increasing interest, with high individual EC exposure levels generally
observed in traffic environments e.g., roadsides (Jia et al., 2021; Zhou et al., 2020). In this study, hourly
EC and $NO_x$ were continuously measured for five years (2016-2020) at a near highway sampling site in
west Shanghai. A machine-learning model i.e., random forest, was applied to train the model to rebuild
the measured EC and $NO_x$ using meteorological and temporal variables as the model input (Grange et
al., 2018; Grange and Carslaw, 2019; Grange et al., 2021; Wang et al., 2022a). The SHAP algorithm
(Lundberg et al., 2020) was used to quantify the impact of meteorological variables on the measured EC
and $NO_x$. A business-as-usual (BAU) scenario was assumed in 2020 and compared with the measured
EC and $NO_x$, quantifying the reduction attributable to the lockdown measures. Implications of future
emission control measures on vehicular emissions are discussed.
**2 Method**
**2.1 Field sampling**
Measurements of the $NO_x$ and EC were conducted continuously from 2016 to 2020 (5 years) at a near
highway sampling site at the Dianshan Lake (DSL) supersite (31.09° N,120.98° E, approximately 15 m
above ground), with two highways (G318 and G50) located approximately 1 km west of the sampling
site. The sampling site is located in Qingpu District in western Shanghai (Fig. S1), 50 km west of
downtown Shanghai. It is at the intersection of Jiangsu, Shanghai, and Zhejiang Provinces. Windrose





analysis showed that the sampling site could be affected by the two nearby highways during both 2016-
2019 (normal years) and 2020 with Covid-19 lockdown measures implemented (Figure S2).
Details of the instrument used to measure EC and $NO_x$ were provided previously (Jia et al., 2020).
Briefly, EC was measured on an hourly basis using a Sunset Carbon Analyzer (Model RT-4, Sunset Lab,
USA), while hourly NO and $NO_2$ were monitored using a Thermo Scientific gas analyzers (Thermo 42i,
Thermo Fisher Scientific, Massachusetts, USA). Meteorological variables of air temperature (air_temp),
wind direction (wd), wind speed (ws), relative humidity (RH), pressure, and rainfall were measured using
a Vaisala automatic weather station (WXT520, Vaisala Ltd., Finland).
Satellite images of $NO_2$ were obtained from Sentinel-5P Level-3 Near Real-Time dataset based on the
observation of the TROPOspheric Monitoring Instrument (TROPOMI) for 2019 and 2020 (Gorelick et
al., 2017). The spatial and temporal distribution of vertical column densities (molecules $cm^{-2}$) of
tropospheric $NO_2$ was used to study the changes in vehicular emissions as a response to strict lockdown
measures implemented in 2020.

**2.2 Data analysis**

**2.2.1 Machine learning set-up and validation**

A machine learning algorithm - Random Forest (Grange et al., 2018; Wang et al., 2022a; Wang et al.,
2022b) was deployed to understand the impact of Covid-19 lockdown on the exhaust emissions from the
near highways in 2020 based on a business as usual (BAU) scenario. $NO_x$ and EC were used as a marker
of traffic exhaust emissions as traffic was its main contributor in Shanghai (Jia et al., 2021). In this study,
the diurnal patterns of EC and $NO_x$ show typical rush hours peaks during both the normal and Covid-19
lockdown periods, consistent with the emission pattern from traffic (Fig. S3).
Meteorological (ws, wd, air_temp, RH, rainfall, and pressure) and time (date_unix, day of the year,
weekday, hour of the day, and day of the lunar year) variables were used as model inputs to explain the
hourly mean EC and $NO_x$ concentrations. The time variable of date_unix is the number of seconds since
1 January 1970. Because the day of the lunar new year is different in the Gregorian calendar, it was
necessary to include the day of the lunar year to better represent the Chinese New Year holiday, which
usually causes a reduction in pollutant concentration during the holiday (Wang et al., 2020; Dai et al.,
2021). For each random forest, the number of trees in the forest was set to 300, while a minimal nod size
was set to five following e (Grange et al., 2018). The training and testing split percentages were 80% and
20% of the dataset, respectively. The random forest model was performed using the latest "rmweather"
R package e (Grange et al., 2018).
Validation of the developed Random Forest was performed by comparing the time series of the
predicted and measured $NO_x$/EC for both the testing and training dataset (Table S1, discussed in Sect.
3.3). The time series of the predicted $NO_x$/EC showed a good agreement with the measured ones with
correlation coefficients in the range of 0.89-0.98 and slopes close to unity, suggesting the developed
Random Forest model captured the variation of the target pollutant well.

**2.2.2 Quantification of the reduction in pollutants attributable to the Covid-19 lockdown**

Based on the developed Random Forest model, the counterfactual $NO_x$ and EC concentrations in a BAU
scenario were derived. The BAU scenario assumed everything was the same in 2020 as in the previous
years. Because the random forest captured the variation of the target pollutant better than the multi-linear



regression model (Table S1), the counterfactual $NO_x$ and EC concentrations reflected the corresponding
pollutant in a BAU scenario better. The long-term measurements of $NO_x$/EC covered multiple years were
necessary to train the model as a comparison to short-term sampling. The BAU analysis was performed
using a function within the "rmweather" R package (Grange et al., 2018).
The counterfactual $NO_x$/EC concentrations were compared with the measured ones during the holiday
(the first week of the lunar year), transition (from day 8 to Lantern Festival, i.e., day 15), and after the
transition period, when the lockdown measures were most restrictive. The differences between the
counterfactual and measured $NO_x$/EC are regarded as the portion that can be attributable to the Covid-
19 lockdown measures (Grange et al., 2021)Specifically, to get the pollutant concentration in a BAU
scenario, a machine learning model was trained by the data over the previous four years to capture the
variability of pollutant concentrations using the same input variables as detailed in Sect. 2.3.1. After
training, the grown forest was used to predict pollutant concentrations experienced beyond the training
period during the Covid-19 lockdown. As a result, the time series of the predicted pollutant beyond the
training period is a counterfactual, representing the model estimation of pollutant concentrations during
the BAU scenario. The pollutant concentrations in the BAU scenario were subsequently compared with
what was observed, with the differences representing the magnitude of the reduction attributable to the
of Covid-19 lockdown.
**2.2.3 Feature importance analysis using the SHAP algorithm**
In this study, SHAP (https://github.com/slundberg/shap) was applied to explain the output of the machine
learning model, quantifying the importance of the meteorological variables (Lundberg et al., 2020;
Oukawa et al., 2022). SHAP is a game theoretic approach that connects optimal credit allocation with
local explanations using the classic Shapley values and their related extensions (Lundberg et al., 2020).
SHAP analysis was performed using the python package of SHAP (version 0.41.0) and scikit-learn
(version 1.2.0).
SHAP produced an interpretable machine-learning model using an additive feature attribution
method (Lundberg et al., 2020). SHAP quantified the contribution of the input meteorological variables
to a single prediction at a specific time, producing a SHAP value in the same unit as the target pollutant.
An overview of which meteorological variables were most important for predicting EC/$NO_x$ was
obtained based on the SHAP values of every feature for every time point. The SHAP overview plot sorted
meteoritical variables by the sum of SHAP value magnitudes over the entire sampling period. SHAP
values were obtained to show the distribution of the impacts each meteorological variables had on the
model output.
**3 Results and Discussion**
**3.1 Trend of observed $NO_x$ during the holiday period and Covid-19 lockdown**
Figure 1a shows the time series of $NO_x$ for 4 weeks before and after the start of the Chinese lunar new
year for 5 years (2016-2020) measurement at the near highway sampling site in west Shanghai (Fig. S1).
To understand the impact of the Covid-19 lockdown measurements on traffic emission, we focus on the
$NO_x$ time series in 2020 in comparison to the averaged time series of $NO_x$ (grey line) for the previous



four years (i.e., the mean of 2016-2019). The beginning of the 2020 lockdown, starting on January 24,
overlapped with the start of the Chinese New Year holiday when human activities have already been
reduced to a large extent as most migrant workers leave the city for their hometowns. Therefore, the
holiday effects need to be taken into account when evaluating the impact of the national lockdown
measures on the measured pollutants at the near highway sampling site.
For 2016-2019, a large reduction in $NO_x$ was seen during the 7-day holiday period when compared to
before the holiday. After the holiday, $NO_x$ levels started to bounce back during the transition period (i.e.,
the period before the lantern festival at DOY 15) and finally reached a similar level after the transition
period when compared to that before the holiday (Fig. 1a).  Specifically, before the holiday, the mean
concentration of $NO_x$ was 72.8 μg m$^{-3}$, while, during the holiday, $NO_x$ concentration was 22.6 μg m$^{-3}$.
After the holiday, the $NO_x$ levels increased from 42.6 μg m$^{-3}$ during the transition to 60.6 μg m$^{-3}$ after
the transition period. Assuming a scenario without the holiday effect, as represented by the arrow line in
Fig. 1b, a reduction of approximately 65% (or 43 μg m$^{-3}$) in the observed $NO_x$ concentration was seen
during the holiday when compared to that before the holiday (72.8 μg m$^{-3}$) for 2016-2019.
Similar to 2016-2019, the observed $NO_x$ in 2020 was also largely reduced (60%) during the holiday
period when compared to before the holiday (Fig. 1b). Specifically, the $NO_x$ before the holiday was 79.5
μg m$^{-3}$, while it was 29.0 μg m$^{-3}$ during the holiday. Because the Covid-19 lockdown started on the same
day as the holiday, the reduction in $NO_x$ observed at the sampling site attributable to the lockdown
measures was smaller than it appeared. In other words, simply comparing the air pollutant concentration
during the first 7-day of lockdown to that before the lockdown would overestimate the impact of Covid-
19 on the measured air pollutant when holiday effects were strong.
However, $NO_x$ remained at low levels during the transition and after the transition period in 2020, i.e.,
the last two weeks during the lockdown, instead of rapidly rising as observed in 2016-2019 (Fig. 1). The
mean concentration during the transition period was 32.6 μg m$^{-3}$ and was 34.8 μg m$^{-3}$ for the last two
weeks during the lockdown in 2020, which was 25% and 50% lower, respectively, when compared to
the same period for 2016-2019. Because it usually takes some time for the control measure to take effect,
focusing on the first 7-day of the lockdown may not represent the true impact of the Covid-19 lockdown
on air quality. Instead, as the lockdown measures took effect, a large reduction in $NO_x$ can be seen at the
late stages of the lockdown when $NO_x$ was supposed to be increasing. Therefore, we focused on the
comparison of $NO_x$ during the last two weeks of the lockdown (labeled as "lockdown" in Fig. 1 and
afterward if not specified otherwise) to study the impact of lockdown measures on traffic emission at
this sampling site.

**3.2 Observed EC reduction attributable to the lockdown control policies**

The measured EC at the near highway sampling site showed a diurnal pattern with a clear morning
rush hour peak, consistent with that for $NO_x$ (Fig. S3), suggesting EC was mainly affected by the nearby
traffic. The measured EC also showed a dependence on wind speed and wind direction, with a higher
concentration associated with low wind speed from the southwest direction, i.e., from the highway (Fig.
S4). The conclusion of EC being mainly from traffic is consistent with previous source apportionment
studies in Shanghai (Chang et al., 2018; Jia et al., 2021).



Figure 2 shows the time series of EC before and during the 2020 lockdown as well as the average time
series of EC (grey line) for the previous four years (i.e., the mean of 2016-2019). Similar to $NO_x$, the
2016-2019 EC level during the holiday was reduced due to the reduced traffic (Fig. 2). Specifically, the
mean EC concentration was 1.08 μg m$^{-3}$ during the holiday, roughly 40% lower compared to that (1.74
μg m$^{-3}$) before the holiday. During the transition period for 2016-2019, EC increased to 1.03 μg m$^{-3}$.
Afterward, EC increased to 1.53 μg m$^{-3}$, very close to the levels before the holiday.
For the 2020 CNY holiday or the first week of the Covid-19 lockdown, EC was also reduced to a
similar level (0.88 μg m$^{-3}$) as 2016-2019 (1.08 μg m$^{-3}$; Fig. 2). Similar to $NO_x$, the EC reduction
attributable to the lockdown measures was not as large as it appeared for the period overlapping with the
holiday. However, EC remained at a low level during (0.92 μg m$^{-3}$) and after the transition (0.78 μg m$^{-3}$)
period. This is because the month-long lockdown measures kept the traffic at a low level for a prolonged
time. This is consistent with the pattern observed for $NO_x$, further confirming the measured EC and $NO_x$
at this near highway sampling site were mainly from traffic emissions. The mean EC concentration
during the transition period or roughly the second week of lockdown in 2020 was 10 % lower than the
same period for 2016-2019, while the mean EC concentration during the last two weeks of lockdown
was 50% lower than the same period for 2016-2019. The low level of EC during and after the transition
period was due to the lockdown measures, reducing the traffic volume and, therefore, reducing the
corresponding traffic-related EC emission.
**3.3 Rebuilding the measured $NO_x$ and EC using a machine learning algorithm**
The measured mass concentrations of atmospheric $NO_x$ and EC were affected by the meteorological
variables including wind speed and wind direction (Fig. S4). This is particularly true for multiple years
of measurement when the meteorological variables varied over these years. Therefore, the concentration
measured at different years was not directly comparable when meteorological variables were varying in
addition to emission strength across years. Moreover, the relationship between the measured $NO_x$/EC
and meteorological conditions was not linear. This is demonstrated by the poor correlation coefficient
(R=0.45-0.48) between the rebuilt $NO_x$/EC and the meteorological parameters using the multilinear
regression model (Table S1). Therefore, the multilinear regression model failed to rebuild the measured
$NO_x$/EC satisfactorily. In this study, the non-linear relationship between $NO_x$/EC and the meteorological
variables was captured by a machine learning algorithm - random forest (See the method section).
Figure 3a shows the scatter plot between the time series of the rebuilt and measured $NO_x$ for the
training and testing dataset.  The predicted $NO_x$ was well correlated with the measured $NO_x$ with a
correlation coefficient (R) of 0.89-0.98, suggesting over 80 % of the data ($R^2$ >0.8) can be explained by
the machine learning model. This value is higher than that from the multilinear regression model (Table
S1). Therefore, the machine learning model demonstrated a better performance than the multilinear
regression model in capturing the relationship between the $NO_x$ and meteorological variables.
Figure 3b shows the scatter plot between the time series of the predicted and measured EC for the
training and testing dataset. Similar to $NO_x$, the rebuilt EC was well correlated with the measured EC
with a correlation coefficient (R) of 0.9-0.98, suggesting over 80 % of the EC can be explained by the
machine learning model. However, for both $NO_x$ and EC, the slope for the linear fit was in the range of



0.67-0.85, suggesting the predicted values were, on average, 13-33% lower than the measured values.
By examining the data, the lower than unity slope was mainly caused by the data points with high
concentrations. These data points can be regarded as outliers that were not captured properly by the
machine learning model since these data points deviated largely from the averaged values.

277       In this study, meteorological variables were used as input variables to train the machine learning model

to rebuild the observed $NO_x$ and EC. However, different meteorological variables had different roles in
affecting the measured $NO_x$ and EC, showing different levels of importance. To evaluate the importance
of different meteorological variables, SHAP model was applied (See method section). Figure 4 shows
the SHAP values (in µg m$^{-3}$) obtained during the rebuilding of $NO_x$ and EC. The meteorological variable
with a high SHAP value was associated with high importance, whereas a SHAP value closer to zero
means the meteorological variable was less important. For $NO_x$, ws is the most important meteorological
variable (Fig. 4), with low ws contributing up to over 100 µg m$^{-3}$ and high ws contributing negatively to
$NO_x$ (down to -40 µg m$^{-3}$). Air temperature, RH, wd, and pressure had SHAP values in the range of -40
µg m$^{-3}$ to 70 µg m$^{-3}$, while rainfall was least important with SHAP values of <10 µg m$^{-3}$ (Fig. 4). Similarly,
ws was also the important variable for EC, with low ws contributing positively to the EC (SHAP value
of up to over 2 µg m$^{-3}$, Fig. 4). Wd, pressure, air temperature, and RH had similar SHAP values (<1.5 µg
m$^{-3}$). Although rainfall was less important, high rainfall was associated with low SHAP values, consistent
with the wet deposition of aerosol.
**3.4 Trend of meteorologically normalized $NO_x$ and EC: a business-as-usual scenario**
To evaluate the impact of the lockdown in 2020 on the $NO_x$/EC emission at this near highway sampling
site, a business-as-usual (BAU) scenario was assumed. The BAU scenario in 2020 assumed that
everything was similar to what would happen previously, i.e., without the lockdown measures. For the
BAU scenario in 2020, $NO_x$ and EC would drop during the holiday, but increase their concentration
levels during the transition and reach a similar level to that before the holiday (Fig. 5), similar to that
observed in 2016-2019 (Fig. 1 and 2). Through the comparison of the 2020 BAU to the measured
$NO_x$/EC in 2020, the reduction in $NO_x$/EC attributable to Covid-19 can be quantitatively evaluated.

299       The $NO_x$ and EC concentrations during the holiday, transition, and lockdown period were normalized

to that before the holiday (Fig. 5). For BAU in 2020, the $NO_x$ during the holiday was reduced to 53% of
the level for that before the holiday. In comparison, the measured $NO_x$ during the holiday was 36% of
the level before the holiday. Therefore, the difference (17%) between BAU-2020 and 2020 was
attributable to the Covid-19 control measures. In other words, the measured $NO_x$ was roughly 30%
(17%/53%) lower than what would be without the control measures. During the transition period, the
$NO_x$ level for BAU-2020 returned to ~75% of the level before the holiday. In comparison, the measured
$NO_x$ was only 40% of that before the holiday. Therefore, the measured $NO_x$ was approximately 45%
lower than the BAU-2020. After the transition period, $NO_x$ returned to a similar level to that before the
holiday for BAU-2020. However, the measured $NO_x$ was only 40% of that before the holiday. As a result,
the $NO_x$ reduction attributable to the Covid-19 lockdown measures was the most significant after the
transition period, which was approximately 60% of the BAU-2020. Therefore, the month-long lockdown





measures kept the NO$_x$ at a low level consistently, demonstrating the effectiveness of the lockdown in
reducing traffic emissions as the lockdown measures continued.

313        Similar to NO$_x$, EC also showed the largest reduction during lockdown when compared to the BAU

2020 (Fig. 5b). Specifically, EC was roughly 60% lower during the lockdown in 2020 than the BAU
scenario in 2020, while the reduction in EC was 40% and 30% lower during the transition and holiday
period, respectively. As a result, both NO$_x$ and EC showed a similar level of reduction which were
attributable to the lockdown measures.

**3.5 Reduction in traffic emission during the Covid-19 lockdown confirmed by satellite monitoring**

Figure 6 shows the TROPOMI images of NO$_2$ in the YRD region over the same period, i.e., before the
holiday and after the transition, for the years 2019 and 2020. By comparing the vertical column densities
of NO$_2$ monitored over the same period in 2019 and 2020, the evolution of satellite-monitoring of NO$_2$
showed a consistent trend with that observed from the ground monitoring at the near highway sampling
site (Fig. 1-3). In particular, a great reduction (50-70%) in NO$_2$ during the lockdown period in 2020 was
seen when compared to that over the same period in 2019, whereas after the transition period in 2020,
NO$_2$ was expected to return to a similar level as that before the holiday i.e., the BAU scenario discussed
in Sect 3.4. Therefore, the reduction (50-70%) in NO$_2$ in 2020 was attributable to the lockdown measures
based on the knowledge gained from the surface monitoring site.

328        Specifically, the vertical column concentration of NO$_2$ at the DSL was highly elevated before the

holiday in 2019 with mean vertical column concentrations of over $18\times10^{15}$ molecules cm$^{-2}$. After the
transition period in 2019, NO$_2$ returned to a slightly lower value ($16\text{-}18\times10^{15}$ molecules cm$^{-2}$) compared
to that before the holiday. This is consistent with BAU scenario assumed in 2020 (Fig. 5). In 2020, NO$_2$
before the holiday was similar to the level over the same period in 2019 ($18\text{-}20\times10^{15}$). However, during
the lockdown period, the NO$_2$ was $8\text{-}10\times10^{15}$, 50-70% lower than in the same period in 2019. Such a
reduction was attributable to the lockdown measures.

**4 Discussion**

336        Through the comparison of EC and NO$_x$ before and during the lockdown in 2020, as well as the same

period in the previous years (2016-2019), we showed that the reduction in vehicular emissions that can
be attributed to the lockdown measures was complicated and cannot be achieved by simply comparing
the concentration difference between before and during the lockdown. This is because vehicular
emissions have their own trend during the Chinese holiday when vehicular emission was largely reduced
(Dai et al., 2021). Here, we show that, due to the overlapping of the first week of lockdown with the
holiday, the reduction in vehicular emission attributable to the lockdown was smaller than it appeared.
This trend can be only revealed from multiple years of continuous measurement and would be easily
missed by a conventional field campaign that only lasted months. This is consistent with the previous
studies (Shi et al., 2021; Dai et al., 2021; He et al., 2020). However, in addition to the holiday effects,
we showed that the reduction in vehicular emission was nearly entirely attributable to the lockdown at a
later stage of lockdown, whereas the holiday and transition period only lasted for 2 weeks.





The lockdown in Shanghai 2020 provided a unique opportunity to study the impact of strict emission
control on local and regional air quality. Many studies have shown the impact of lockdown on traffic
emission, but with different degrees of impact partly because the duration of the lockdown was month-
long and partly overlapped with the holiday as shown in this study (Jia et al., 2020; Dai et al., 2021; Shi
et al., 2021; Wang et al., 2020). However, most previous studies focused on gas pollutant i.e., $NO_2$
probably because $NO_2$ was a regular gas pollutant that is routinely measured at the air quality monitoring
sites across the major Chinese cities (He et al., 2020), while few reported the particulate EC emission
from traffic partly due to the scarcity of the dataset. EC is light absorbing and is regarded as a warming
agent second to $CO_2$ (Cappa et al., 2012; Jacobson, 2001; Liu et al., 2015). In addition, EC is one of the
major particulate pollutants that can cause adverse health effects (Daellenbach et al., 2020; Rappazzo et
al., 2015).  To the best of our knowledge, this is the first study to illustrate the impact of lockdown on
vehicular EC emissions at a near highway sampling site based on 5-years of continuous measurement.
Such a dataset is rare in the literature since lockdown measures restrict the movement of instrument
operators. Only with good maintenance of the instrument at the sampling site can we keep the sampling
going on during the strict lockdown.
To decouple the effects of the meteorological variables on the measured $NO_x$ and EC, a machine
learning model was trained and tested based on the 5-year dataset. The machine learning model emerges
as a powerful model in air quality studies especially the development of SHAP (Lundberg et al., 2020)
making the machine learning model explainable rather than a black box as in most previous air quality
studies (Grange and Carslaw, 2019; Grange et al., 2017; Shi et al., 2021; Vu et al., 2019). The explainable
machine learning model of SHAP showed meteorological variables especially ws and wd were key
parameters that affect the measured levels with concentrations of up to 100 μg m$^{-3}$ for $NO_x$. Due to
important the role of meteorological variables, their impact needs to be removed when evaluating the
true impact of the lockdown on vehicular emissions. Here, instead of simply comparing the concentration
before and during the lockdown, a BAU scenario was assumed in 2020. This relies on the rebuilding
power of the mathematical model. However, to train the machine learning model, a large body of datasets
is required as input. As more datasets are to be collected and used as model input, the performance of
machine learning is expected to improve further. Moreover, with more variables, e.g., vehicular types,
weight, and road conditions, being monitored and used as input for the model, a better prediction power
of the machine learning is anticipated. Correspondingly, the air quality improvement upon future
emission control scenarios can be better predicted.
**5 Conclusion**
In this study, the time series of vehicular emissions of $NO_x$ and EC before and during the 2020 lockdown
as well as the averaged time series of $NO_x$ over the same period for the previous four years (i.e., the mean
of 2016-2019) were compared and used to train the machine learning model, rebuilding the $NO_x$ and EC
in a BAU scenario in 2020. Meteorological variables especially wind speed and direction were found to
be the key parameters that affect the measured levels with concentrations of up to 100 μg m$^{-3}$ for $NO_x$
using the explainable machine learning model of SHAP. Due to important the role of meteorological
variables, their impact needs to be removed when evaluating the true impact of the lockdown on vehicular





emissions. In contrast, by simply comparing the concentration before and during the lockdown, the effects of the lockdown on air pollutant emission can be misrepresented. The results show that vehicular emissions had their own trend during the Chinese holiday during which vehicular emission was largely reduced. Because the first week of lockdown overlapped with the holiday, the reduction in vehicular emissions attributable to the lockdown was smaller than it appeared. This is in line with previous studies that took into account the holiday effects using a machine learning based de-weathering process. However, different from previous studies, a large reduction (50-70%) in vehicular emissions of $NO_x$ and EC was attributed to the lockdown at a later stage. This value is larger than previous studies because both the holiday effects and meteorological impacts were removed during this period. This large reduction in vehicular emissions at a later stage was confirmed by satellite monitoring of $NO_2$. Therefore, strict lockdown reduced both vehicular gaseous and particulate emission significantly when holiday and meteorological effects were not affecting the trend analysis. This study demonstrates the importance of continuous monitoring at this Shanghai supersite. When coupled with an advanced mathematical algorithm, insights into the impact of human activities on air pollution can be gained based on long-term monitoring. Air quality improvement in future emission control scenarios is expected to be better predicted.

**Associate content**

Supporting Information
Supplementary figures (Fig. S1-S4).

**Credit authorship contribution statement**

MW, ZZ, XL and SH designed the study. YD, JH, JC, YL and QF conducted field campaign. MW, YD, ZZ and QY conducted data analysis. MW prepared the manuscript with contributions from all co-authors. QF, TW, JC and SL provided input for revision before submission. QF and SL provided project guidance.

**Declaration of competing interest**

The authors declare that they have no conflicting interests.

**Acknowledgements**

This work was supported by the Start-up Fund for RAPs under the Strategic Hiring Scheme (P0043854), Green Tech Fund (GTF202110151), Environment and Conservation Fund-Environmental Research, Technology Demonstration and Conference Projects (ECF 63/2019), the RGC Theme-based Research Scheme (T24-504/17-N), the RGC Theme-based Research Scheme (T31-603/21-N), Key Research and Development Projects of Shanghai Science and Technology Commission (20dz1204000), State Ecology and Environment Scientific Observation and Research Station for the Yangtze River Delta at Dianshan Lake (SEED).



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

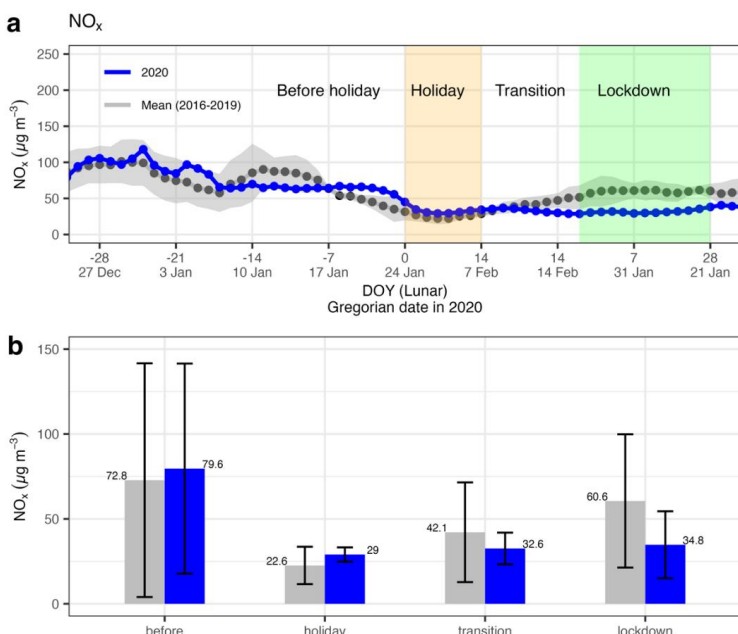

**Figure 1. (a) Time series (day of the year; DOY) of the measured NO$_x$ for 4 weeks before and after the start of the Chinese Lunar year for the mean of 2016-2019 and 2020; and (b) Mean NO$_x$ concentrations for different periods, i.e., before the holiday, holiday, transition and lockdown. The time series in (a) was a 7-day rolling average. The error bar in (b) stands for one standard deviation. Note that the lunar DOY for 2016-2019 was on different Gregorian date, but were grouped together based on lunar DOY in (a).**



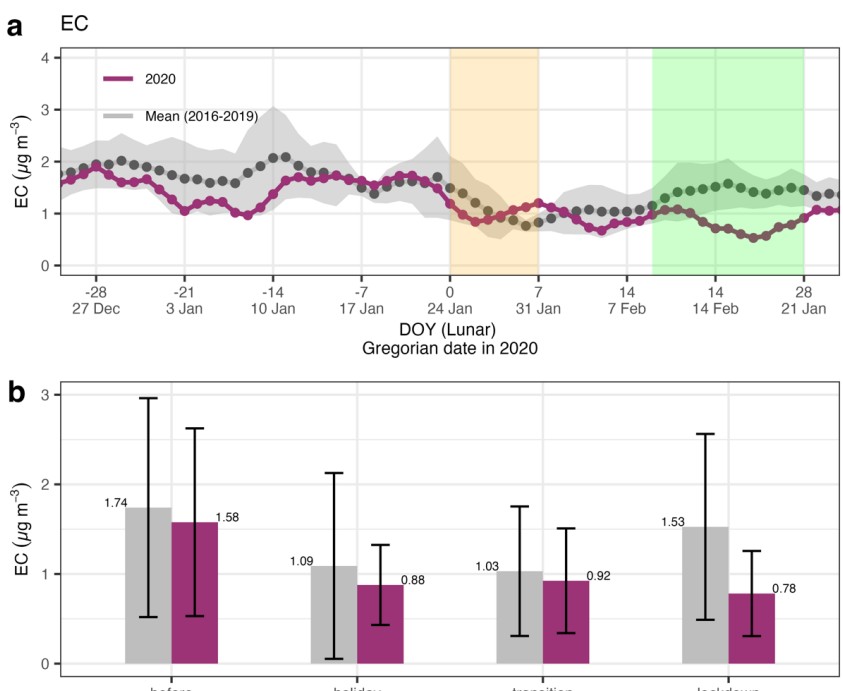

Figure 2. (a) Time series (day of the year; DOY) of the measured EC for 4 weeks before and after the start of the Chinese Lunar year for the mean of 2016-2019 and 2020; and (b) Mean EC concentrations for different periods, i.e., before the holiday, holiday, transition and lockdown. The time series in (a) was a 7-day rolling average. The error bar in (b) stands for one standard deviation.



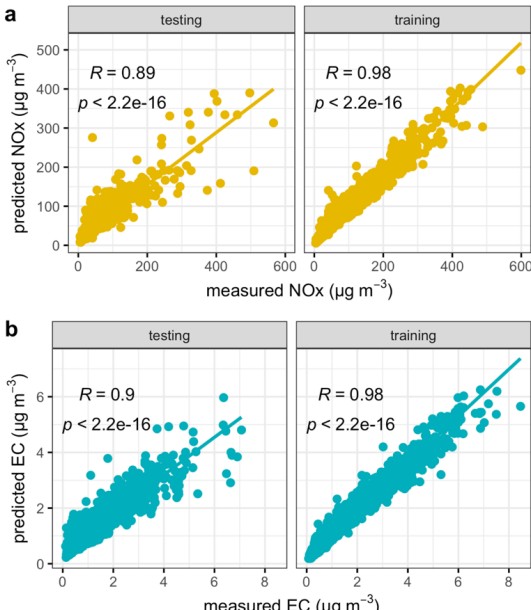

**Figure 3. Scatter plot between the predicted and measured (a) NO$_x$ and (b) EC for the testing and training dataset. Also shown is the linear regression between the predicted and measured values, with the correlation coefficient (R) and p-value in the top left.**



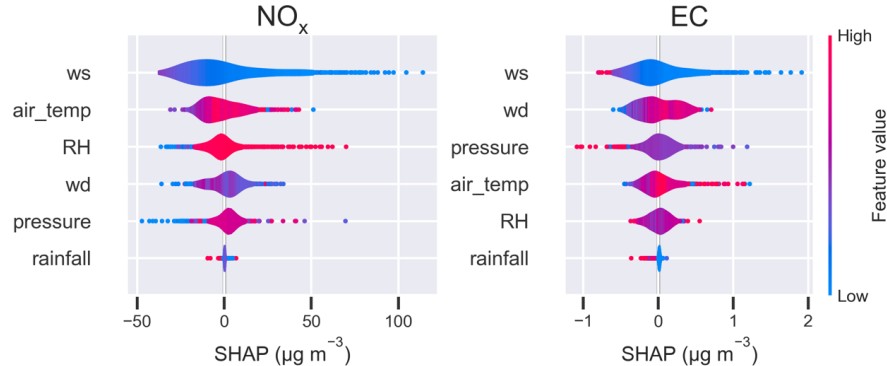

**Figure 4. SHAP values (in μg m⁻³) for the meteorological variables i.e., features when building the random forest model for NOₓ and EC.**



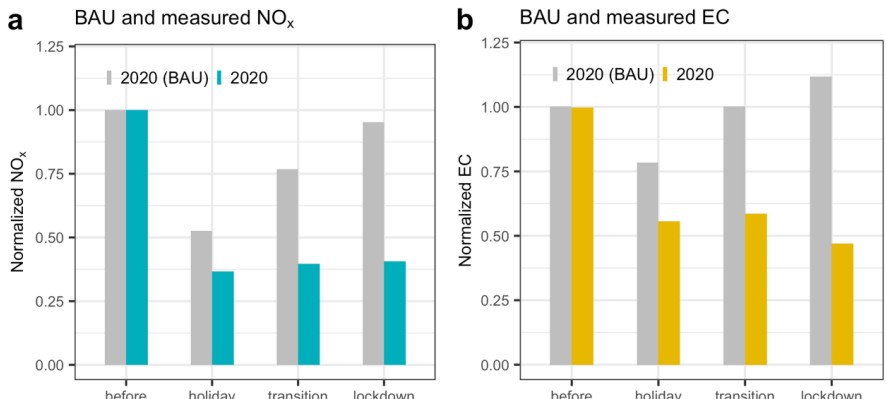

**Figure 5. Comparison of NO$_x$ (a) and EC (b) evolution between the business-as-usual (BAU) scenario and the measured one in 2020. All concentrations were normalized to the level before the holiday.**

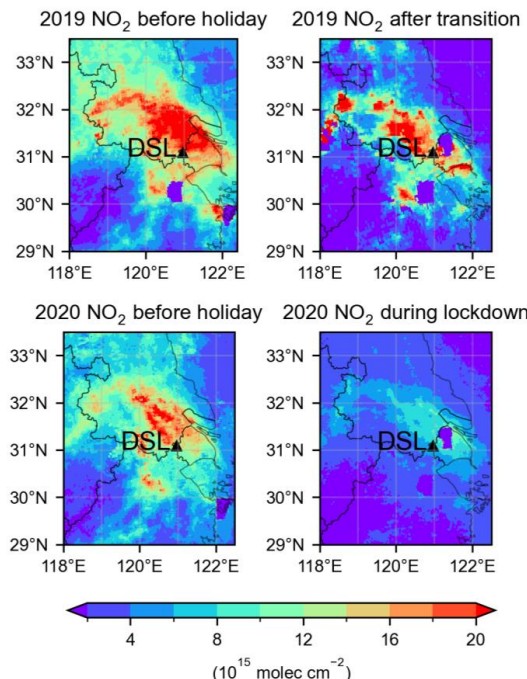

**Figure 6. The spatial distribution of TROPOMI NO₂ over the same period in 2019 and 2020 near the DSL sampling site in west Shanghai in the YRD region.**