# Peer review of "Covid-19 lockdown in Shanghai: insights from 5-year"

_EGUsphere, 2023_

## Referee Comment (RC1)

**General comments on the paper entitled "Reduction in vehicular emissions attributable to the Covid-19 lockdown in Shanghai: insights from 5-year monitoring-based machine learning":**

This work presents the results of a 5-year monitoring of EC and $NO_x$ in a traffic site in Shanghai, China. The authors used Random Forest to estimate a business-as-usual scenario during the COVID-19 lockdown period. The authors have validated their results using Satellite data of $NO_2$.

This manuscript can be interesting and is generally well-written. However, there are a lot of important clarifications that need to be addressed especially in the methodology. It is necessary to improve the manuscript through more elaborate discussions and concise take-aways in line with the results presented in this manuscript.

General comments:

- Line 86: Please put a period after the sentence.
- Line 95 to 101: This paragraph is more appropriate in Line 1. Please bear in mind the cohesiveness on the next paragraphs.
- Introduction: Please include more recent studies similar to your work. There are quite a few on COVID-19 lockdown implications and traffic and some have similarly estimated a business-as-usual scenario. Here are a few of those: https://doi.org/10.1039/D3EA00013C, https://doi.org/10.1007/s11869-023-01330-3, https://doi.org/10.1080/02786826.2023.2193237.
- Line 122: Please include the units of each of the meteorological variables. What are the resolution of these observations?
- Line 124: What is the expected seasonality of the air quality parameters considered in the study area? Please elaborate on the seasonal trends of both EC and $NO_x$.
- Line 132: Was it Random Forest regression that was specifically used in this study? A modelling workflow would be useful to clearly present the Random Forest methodology.
- Line 138: Did you test on any other meteorological variable aside from the mentioned variables? Were all these variables/features included in the final Random Forest model? If yes, what was the criteria used to include/exclude features in the model?
- Line 138: Please elaborate on the units used for the wind direction feature. Did you perform any specific data processing on this variable or in any of the other features?
- Line 132: What is the size of the dataframe worked on this study? How many data points in total was used to train and test the model? How many data points were available for both target and features in the Random Forest? How many data points are available in the 3 periods that were compared in this study (during, transition, lockdown)?
- Line 144: How did you determine that this was the appropriate architecture for the Random Forest model in this study? Was the training and test sets partitioned randomly?
- Line 144: Please check if there is an existing decreasing trend for both EC and $NO_x$ in the years prior to 2020. If there is, it could be useful to use the first 80% of the dataset as the training set and the last 20% as the testing set. In this case, the existing trends can be taken into consideration in the building of the Random Forest model.
- Line 148: What are the model performance measures used? In this paragraph, it was mentioned that a validation step was performed by comparing the predicted and measured ratio of $NO_x$ and EC in the training and testing step. Please elaborate on why this methodology was used to validate the results. There are many different ways to validate a machine learning model. Have you performed an out-of-bag model validation

(using a subset of your dataset not included in both training and testing sets)? The validation step of any machine learning model has to be clear and elaborate to support that the model optimization was reached and substantial. This was not apparent in the methodology of this study.

- Line 157: I suggest the use the term "estimated" than "counterfactual".
- Line 165: Please add a period after the sentence.
- Line172: How did you calculate the differences representing the magnitude of reduction? In the results and discussion section, this difference is in a percentage form. Please be clear on how this was calculated? Is this percentage difference, percentage change? Was this difference calculated on a daily/hourly comparison or using the entire lockdown period? Please explain.
- Line 204: Please add the standard deviation on the mean concentration. Please apply this on all other concentrations mentioned in this section.
- Line 207: This sentence is confusing, please rephrase. Did you mean a 65% reduction was found and that is equivalent to a 43 µg m$^{-3}$ reduction?
- Line 223: I understand that the effect of the holidays can indeed affect the overall reduction of pollution levels, hence it does make sense to only use the last two weeks of the lockdown period. However, since the holiday is a yearly event, it would still be interesting to see the reduction of levels during the entire lockdown period even if it does include the holidays. This should be included in the analysis as well.
- Line 258: What is the correlation coefficient mentioned here? Is this Pearson or Spearman or something else? Please include the RSQ value as done in Line 265.
- Line 260: The term failed can be subjective as some studies have referred to this level of correlation as moderate. Please rephrase.
- Line 261: This sentence is quite redundant as this has been mentioned several times in the prior sections already. Please check other redundant statements in the manuscript.
- Line 271: The RSQ value is missing. Is this also 80% same as for NO$_x$?
- Line 277 to 290: Does the feature importance (SHAP values) make sense? In theory, temperature should probably have been a strong feature for NO$_x$, but it doesn't seem to be. Why do you think that is? What is the wind speed considered in this study, is this wind speed close to the ground? There is a need to elaborate the seasonal/climate conditions during the lockdown restrictions in the study area? What are the specific (for example) seasonal influence on the levels of NO$_x$ and EC?
- Line 283: In this sentence, it was mentioned that low wind speed contributes up to 100 µg m$^{-3}$ in NO$_x$ and high wind speed contributes negatively. However, Figure 4 presents that wind speed did not vary a lot. How did the authors categorize low and high wind speed in this analysis? A lot of the SHAP values are resting in zero value, which could mean very little impact any of the features. How about the temporal variables considered What is the impact of these variables?
- Section 3.5: The use of Satellite data to validate the results does not add much information in the analysis. Usually ground measurements are used to validate Satellite data. Hence, this analysis does not make sense and seems out of place.
- Line 341: Please check the tenses of the verbs used in the manuscript. Check for the rest of the text.
- Line 358: There are already existing studies focusing on traffic emissions, which also uses long-term datasets and using Random Forest to estimate BAU levels. There is a need to discuss/compare the results in this study with existing literature.
- Line 367: Please explain the geochemical meaning behind high importance of wind speed and wind direction as features in a Random Forest model.

- Section 4: The discussion section lacks elaborate discussions about the results of this study. How does the results of this study compared to other studies using a more traditional/simplistic approach on evaluating the effects of the lockdown restrictions on air pollution? Is there an under- or over-estimation in other approaches compared to using a Random Forest model? Since the measurements are in an hourly resolution, was there any diurnal or hourly variation during the lockdown period?
- Section 5: The conclusions section read like a summary. This section needs to be improved. I suggest using bullet points to enumerate the main take-aways of the study.

Additional comments:

- Was it only vehicle/transportation that was restricted in Shanghai? How about the industrial sector? There are other sources of $NO_x$ and EC apart from vehicular emissions. Are there no other known anthropogenic sources in the area?
- Was there an already existing trend that needs to be considered apart from the impact of COVID-19 restrictions? A long-term dataset should be able to check this.
- There is a lot about the % reduction but the analysis and discussions lack on the model optimization. Can we use SHAP to improve the Random Forest model by choosing the most appropriate features to be used in the model?
- Figure 1a and Figure 2a both show time-series of $NO_x$ and EC. Is this a daily average? If it is, please add the standard deviation for year 2020. This might show that in fact the change is not as apparent as it seems.
- It is essential that a dependence scatter plot be provided to show the effect of a single feature across the whole dataset. It will also be useful to just take the mean absolute value of the SHAP values for each feature to get a standard bar plot that can be easily interpreted.

---

## Author Response (AR1)

We thank the reviewers for their comments that helped improve the manuscript. We have now revised the manuscript accordingly. Please find the point-to-point response to each comment below, where our responses are in blue and the revisions in the main text are in red.

Reviewer #1

General comments on the paper entitled "Reduction in vehicular emissions attributable to the Covid-19 lockdown in Shanghai: insights from 5-year monitoring-based machine learning":

This work presents the results of a 5-year monitoring of EC and NOx in a traffic site in Shanghai, China. The authors used Random Forest to estimate a business-as-usual scenario during the COVID-19 lockdown period. The authors have validated their results using Satellite data of NO2. This manuscript can be interesting and is generally well-written. However, there are a lot of important clarifications that need to be addressed especially in the methodology. It is necessary to improve the manuscript through more elaborate discussions and concise take-aways in line with the results presented in this manuscript.

Response: We thank the reviewers for his/her careful comments. We have provided more clarifications in the methodology section. We have provided more elaborate discussions and concise take-aways. Please see the point-to-point response below.

General comments:

Line 86: Please put a period after the sentence.

Response: corrected.

Line 95 to 101: This paragraph is more appropriate in Line 1. Please bear in mind the cohesiveness on the next paragraphs.

Response: We have now moved this paragraph to the start of the discussion and taken into consideration of cohesiveness.

Introduction: Please include more recent studies similar to your work. There are quite a few on COVID-19 lockdown implications and traffic and some have similarly estimated a business-as-usual scenario. Here are a few of those:

https://doi.org/10.1039/D3EA00013C, https://doi.org/10.1007/s11869-023-01330-3, https://doi.org/10.1080/02786826.2023.2193237.

Response: we have now cited more recent studies on Covid-19 lockdown implications and traffic, including the 3 mentioned above and (González-Pardo et al., 2022).

Line 122: Please include the units of each of the meteorological variables. What are the resolution of these observations?

Response: We have included units of the meteorological variables. The time resolution of these observations is 1 hour. It read, "…Meteorological variables of air temperature (air_temp; $^{\circ}$C), wind direction (wd; degree), wind speed (ws; m s$^{-1}$), relative humidity (RH; %), pressure (hPa), and rainfall (mm) were measured using a Vaisala automatic weather station (WXT520, Vaisala Ltd., Finland) with a time resolution of 1 hour…"

Line 124: What is the expected seasonality of the air quality parameters considered in the study area? Please elaborate on the seasonal trends of both EC and NOx.

Response: We have now elaborated on the seasonal trends of both EC and NOx and have added Figure S3. It now reads, "…The seasonal variation of EC and $NO_x$ is shown in Figure S3. For 2015-2019, the median of EC varied in the range of 1.0-1.5 µg m$^{-3}$ with higher concentrations in winter than in summer. The median of $NO_x$ varied in the range of 45-55 µg m$^{-3}$ with higher concentrations in winter than in summer for 2015-2019. The Covid-19 lockdown measures were implemented in 2020, resulting in lower concentrations of $NO_x$/EC but similar seasonal trend (Figure S3)…"

**Figure S3. Seasonal variation of EC and NOx for 2016-2019 and 2020.**

Line 132: Was it Random Forest regression that was specifically used in this study? A modelling workflow would be useful to clearly present the Random Forest methodology.

Response: Random Forest has been applied in modelling air pollutants in previous studies (Grange et al., 2021; Wang et al., 2022a). Here, Random Forest was applied to model the EC and NOx to study the Covid-19 impact on traffic emissions, the first of its kind in Shanghai. A modelling workflow is now added as Figure S4.

[Figure]

**Figure S4. Flow chart of the method.**

Line 138: Did you test on any other meteorological variable aside from the mentioned variables? Were all these variables/features included in the final Random Forest model? If yes, what was the criteria used to include/exclude features in the model?

Response: We did not test other variables because only these meteorological variables were measured at the same site. All these variables were included because all features were potential factors that could influence the measured concentrations (Qin et al., 2022; Wang et al., 2022a). There was a test of feature importance in Figure 4. All these features were included with different levels of relative importance.

Line 138: Please elaborate on the units used for the wind direction feature. Did you perform any specific data processing on this variable or in any of the other features?

Response: The unit of wind direction is degree (please also see the response to the previous comment). No pre-processing was performed on wind direction or other meteorological variables.

Line 132: What is the size of the dataframe worked on this study? How many data points in total was used to train and test the model? How many data points were available for both target and features in the Random Forest? How many data points are available in the 3 periods that were compared in this study (during, transition, lockdown)?

Response: We have now provided more details about the size of the data frame. In the revised

Method section 2.2.1 and 2.2.2, It now reads, "…The time resolution for the random forest features and the target was 1 hour. The Covid-19 lockdown started in late January 2020 and lasted roughly 1 month (see Fig. 1). The number of data points modelled in the Random Forest model was 6244, covering one month before and after the start of Covid-19 lockdown for the same period for 5 years (Fig. 1). Data with missing values were excluded (8% of the data). Data before the start of the Lunar new year (i.e., January 24, 2020) were used to train and test the model with a total number of data points of 5616.   80% (4493 data points) of the dataset was randomly selected to train the dataset, while the rest 20% (1123 data points) of the dataset was used to test the model. The training-testing percentages followed Grange et al. (2021). The random forest model was performed using the latest "rmweather" R package e (Grange et al., 2018). Based on the built forest, data after the Lunar new year was estimated using the features during the Covid-19 period, i.e., the BAU scenario (Fig. S4)…"

Also "…The estimated $NO_x$/EC concentrations were compared with the measured ones during the holiday (the first week of the lunar year, 167 data points), transition (from day 8 to Lantern Festival, i.e., day 15; 206 data points), and after the transition period (250 data points), when the lockdown measures were most restrictive …"

Line 144: How did you determine that this was the appropriate architecture for the Random Forest model in this study? Was the training and test sets partitioned randomly?

Response: The training and test sets were randomly selected with model set-up followed Grange et al. (2021). The training and testing datasets were portioned randomly (also see the reply to the previous comment).

In the text, it now reads, "…80% (4493 data points) of the dataset was randomly selected to train the dataset, while the rest 20% (1123 data points) of the dataset was used to test the model. The training-testing percentages followed Grange et al. (2021)…"

Line 144: Please check if there is an existing decreasing trend for both EC and NOx in the years prior to 2020. If there is, it could be useful to use the first 80% of the dataset as the training set and the last 20% as the testing set. In this case, the existing trends can be taken into consideration in the building of the Random Forest model.

Response: The trend was taken into account by the model because the time variable of date_unix was used. The time variable of date_unix is the number of seconds since 1 January 1970 (Grange et al., 2018).

Line 148: What are the model performance measures used? In this paragraph, it was mentioned that a validation step was performed by comparing the predicted and measured ratio of NOx and EC in the training and testing step. Please elaborate on why this methodology was used to validate the results. There are many different ways to validate a machine learning model. Have you performed an out-of-bag model validation (using a subset of your dataset not included in both training and testing sets)? The validation step of any machine learning model has to be clear and elaborate to support that the model optimization was reached and substantial. This was not apparent in the methodology of this study.

Response: We agree that there are many different ways to validate the model. Here, the model performance was measured based on correlation coefficient R and slope between the

measured and predicted pollutant (see Figure 3 and Table S1). The correlation coefficient is often used to measure the performance of the random forest model (González-Pardo et al., 2022). The model set-up followed Grange et al. (2021) and showed a good performance with R in the range of 0.89-0.98 for the training and testing dataset (Figure 3 and Table S1). We have elaborated on how the model performance was measured in the revised text.

In the revised Sect. 2.2.1, it now reads, "…Validation of the developed Random Forest was performed by comparing the time series of the predicted and measured $NO_x/EC$ for both the testing and training dataset based on the correlation coefficient R and slope between the time series of measured and predicted pollutant. A good simulation often features a high value of correlation coefficient (>0.6) and slope close to unity (Grange et al., 2021; González-Pardo et al., 2022; Qin et al., 2022)..."

Line 157: I suggest the use the term "estimated" than "counterfactual".
Response: The term "estimated" is now used as suggested.

Line 165: Please add a period after the sentence.
Response: corrected.

Line172: How did you calculate the differences representing the magnitude of reduction? In the results and discussion section, this difference is in a percentage form. Please be clear on how this was calculated? Is this percentage difference, percentage change? Was this difference calculated on a daily/hourly comparison or using the entire lockdown period? Please explain.
Response: The differences representing the magnitude of reduction were calculated based on the comparison to the BAU scenario in 2020 (see the workflow; Figure S4). This is in a percentage form using the entire period. We have now provided a workflow (Figure S4) on how this was calculated.

Line 204: Please add the standard deviation on the mean concentration. Please apply this on all other concentrations mentioned in this section.
Response: Now added.

Line 207: This sentence is confusing, please rephrase. Did you mean a 65% reduction was found and that is equivalent to a 43 µg m-3 reduction?
Response: We have now rephrased this sentence. 65% reduction was equivalent to 50.2 µg m$^{-3}$. In the revised Sect. 3.1, it now reads, "…As a result, compared to the average $NO_x$ level (72.8 µg m$^{-3}$) before the holiday, $NO_x$ was reduced by over 65% (i.e., 50.2 µg m$^{-3}$) during the holiday for a normal year…"

Line 223: I understand that the effect of the holidays can indeed affect the overall reduction of pollution levels, hence it does make sense to only use the last two weeks of the lockdown period. However, since the holiday is a yearly event, it would still be interesting to see the reduction of levels during the entire lockdown period even if it does include the holidays. This should be included in the analysis as well.

Response: We agree that it is interesting to see the reduction during the entire lockdown period. Based on comparison with the BAU analysis, Figure 6 shows the reduction over the holiday, transition, and lockdown period.

Line 258: What is the correlation coefficient mentioned here? Is this Pearson or Spearman or something else? Please include the RSQ value as done in Line 265.

Response: It is Pearson. RSQ is now added. It now reads, "…This is demonstrated by the relatively low values of correlation coefficient (i.e., Pearson's R of 0.45-0.48 or $R^2$ of 0.20-0.23) between the rebuilt $NO_x$/EC and the meteorological parameters using the multilinear regression model (Table S1)…"

Line 260: The term failed can be subjective as some studies have referred to this level of correlation as moderate. Please rephrase.

Response: We have removed the term 'poor' and rephrased this sentence to "…This is demonstrated by the relatively low values of correlation coefficient (i.e., Pearson's R of 0.45-0.48 or $R^2$ of 0.20-0.23) between the rebuilt $NO_x$/EC and the meteorological parameters using the multilinear regression model (Table S1)…"

Line 261: This sentence is quite redundant as this has been mentioned several times in the prior sections already. Please check other redundant statements in the manuscript.

Response: The redundant sentences are now removed here and elsewhere.

Line 271: The RSQ value is missing. Is this also 80% same as for NOx?

Response: RSQ is now added. RSQ values are similar for NOx.

Line 277 to 290: Does the feature importance (SHAP values) make sense? In theory, temperature should probably have been a strong feature for NOx , but it doesn't seem to be. Why do you think that is? What is the wind speed considered in this study, is this wind speed close to the ground? There is a need to elaborate the seasonal/climate conditions during the lockdown restrictions in the study area? What are the specific (for example) seasonal influence on the levels of NOx and EC?

Response: We agree that temperatures, in theory, have a strong impact on NOx emissions. The SHAP modeled the importance of features (e.g., wind speed, wind direction, and temperatures) when predicting NOx. The SHAP modelling result shows that temperature was the second most important feature in predicting NOx after wind speed. This is probably because, after emissions, NOx was subject to dilution during the transport from the emission source (i.e., traffic) to the sampling site. Low wind speed indicated poor dispersion conditions that favored the build-up of air pollutants.

The wind speed is close to the ground (15 m above the ground), which was mentioned in the method section.

The seasonal variation of NOx and EC is shown in Fig. S3 (see the response to the previous comment). The lockdown period was in winter and the BAU scenario was assumed in winter, and, therefore, the seasonal variations were expected to be a minor influence. A more detailed

discussion about seasonal influences is available in Wang et al. (2022b).

Line 283: In this sentence, it was mentioned that low wind speed contributes up to 100 µg m-3 in NOx and high wind speed contributes negatively. However, Figure 4 presents that wind speed did not vary a lot. How did the authors categorize low and high wind speed in this analysis? A lot of the SHAP values are resting in zero value, which could mean very little impact any of the features. How about the temporal variables considered. What is the impact of these variables?
Response: Figure 4 shows the distribution of the SHAP values from the SHAP modelling, color-coded by the magnitude of the features (from low to high quantiles). Figure 4 shows, for certain datapoint, wind speed contributed up to 100 µg m$^{-3}$ of NOx. For most other modelled data points, the importance of wind speed in predicting NOx was lower, i.e., close to zero. The low and high wind speed is based on the distribution of wind speed based on quantiles.
We have now added the mean absolute SHAP value in the revised Figure 4. SHAP model only considered meteorological variables.

[Figure]

**Figure 4. Distribution of SHAP values (in µg m$^{-3}$) for the meteorological variables i.e., features when building the random forest model for NOₓ (a) and EC (b); and mean absolute SHAP values for NOₓ (c) and EC (d).**

Section 3.5: The use of Satellite data to validate the results does not add much information in the analysis. Usually ground measurements are used to validate Satellite data. Hence, this

analysis does not make sense and seems out of place.

Response: We agree that the original description of satellite data can cause confusion. While our near roadside sampling site provided information on the current receptor site, satellite images provided regional information on the impact of Covid-19 on traffic emissions. We changed the title of Sect 3.5 to "Reduction in traffic emission during the Covid-19 lockdown on a regional scale".

Line 341: Please check the tenses of the verbs used in the manuscript. Check for the rest of the text.

Response: Checked.

Line 358: There are already existing studies focusing on traffic emissions, which also uses long-term datasets and using Random Forest to estimate BAU levels. There is a need to discuss/compare the results in this study with existing literature.

Response: We agree that there are existing studies focusing on NOx emissions from traffic, but studies on EC emissions are rare. We have discussed this with existing literature in this section. It now reads, "…For example, Jia et al. (2020) reported a 56-58% reduction in $NO_x$ during the Covid-19 lockdown period by directly comparing the NOx concentrations to the before-holiday period in Shanghai. Here, we showed NOx was already reduced by approximately 60% during the holiday week for a normal year. Such a trend in traffic emissions during the holiday week is consistent with the findings from previous studies (He et al., 2020; Dai et al., 2021; Shi et al., 2021). Considering the holiday effect, Dai et al. (2021) reported a reduction of ~15% in NO2 attributable to the Covid-19 lockdown period in Shanghai during the holiday week. This value is similar to this study's 17% reduction in NOx. However, previous studies focusing on only the holiday week may underestimate the impact of the Covid-19 lockdown on air quality over an extended period because the holiday period lasted more than one week. During the last two weeks of the lockdown, an approximately 60% reduction in both NOx and EC was attributable to the Covid-19 lockdown. Since the lockdown measures often take time to be executed more extensively, the later stages of air pollution reduction may better represent the air quality effect of Covid-19…"

Line 367: Please explain the geochemical meaning behind high importance of wind speed and wind direction as features in a Random Forest model.

Response: Low wind speed indicated poor dispersion conditions, which favored the build-up of air pollutants from traffic emissions. We have provided a polar plot of EC and NOx as a function of wind speed and wind direction in Fig. S6. It now reads, "…Low wind speed was indicating poor dispersion conditions that favored the build-up of air pollutants, while wind direction pointed to the emission source from nearby traffic.…"

Section 4: The discussion section lacks elaborate discussions about the results of this study. How does the results of this study compared to other studies using a more traditional/simplistic approach on evaluating the effects of the lockdown restrictions on air pollution? Is there an under- or over-estimation in other approaches compared to using a Random Forest model? Since the measurements are in an hourly resolution, was there any diurnal or hourly variation

during the lockdown period?

Response: We have now added a more elaborate discussion about the results from this study and compared it to previous studies using Random Forest. Diurnal variations were provided in Fig. S5.

In the revised Sect. 4, it now reads, "…For example, Jia et al. (2020) reported a 56-58% reduction in $NO_x$ during the Covid-19 lockdown period by directly comparing the NOx concentrations to the before-holiday period in Shanghai. Here, we showed NOx was already reduced by approximately 60% during the holiday week for a normal year. Such a trend in traffic emissions during the holiday week is consistent with the findings from previous studies (He et al., 2020; Dai et al., 2021; Shi et al., 2021). Considering the holiday effect, 字段 Dai et al. (2021) reported a reduction of ~15% in NO2 attributable to the Covid-19 lockdown period in Shanghai during the holiday week. This value is similar to this study's 17% reduction in NOx.   However, previous studies focusing on only the holiday week may underestimate the impact of the Covid-19 lockdown on air quality over an extended period because the holiday period lasted more than one week. During the last two weeks of the lockdown, an approximately 60% reduction in both NOx and EC was attributable to the Covid-19 lockdown. Since the lockdown measures often take time to be executed more extensively, the later stages of air pollution reduction may better represent the air quality effect of Covid-19…"

Section 5: The conclusions section read like a summary. This section needs to be improved. I suggest using bullet points to enumerate the main take-aways of the study.

Response: We have improved this section and used bullet points to enumerate the main take-aways of this study. It now reads, "…In this study, we studied the impact of the Covid-19 lockdown on traffic emissions based on a 5-year measurement of NOx and EC using a BAU scenario analysis at a near highway sampling site in Shanghai. We showed that 1) by simply comparing the concentration before and during the lockdown, the effects of the lockdown on air pollutant emission may be over-estimated; 2) a large reduction (50-70%) in vehicular emissions of $NO_x$ and EC was attributed to the lockdown at a later stage that may better represent the impact of lockdown measures on air quality. This value is larger than previous studies because both the holiday effects and meteorological impacts were removed during this period. This large reduction in vehicular emissions at a later stage was consistent with satellite monitoring of $NO_2$.…"

Additional comments:

Was it only vehicle/transportation that was restricted in Shanghai? How about the industrial sector? There are other sources of NOx and EC apart from vehicular emissions. Are there no other known anthropogenic sources in the area?

Response: Vehicle/transportation are the major source of NOx and EC that were monitored at the near highway sampling site. This is demonstrated by the diurnal pattern that showed a typical morning rush hour peak. The conclusion of EC/NOx being mainly from traffic is consistent with previous source apportionment studies in Shanghai (Chang et al., 2018; Jia et al., 2021). These discussions were in the main text.

Was there an already existing trend that needs to be considered apart from the impact of

COVID-19 restrictions? A long-term dataset should be able to check this.

Response: Please see the response to the previous comment above on the seasonal trend of air pollutants.

There is a lot about the % reduction but the analysis and discussions lack on the model optimization. Can we use SHAP to improve the Random Forest model by choosing the most appropriate features to be used in the model?

Response: The current model set-up was trained and tested using our datasets. It demonstrated a good performance (please see the reply to the previous comment).

SHAP was used to explain the model importance of the features. Features with high importance are important input variables. However, features with relatively low importance (e.g., precipitation) are also included since wet deposition was a key atmospheric process. We keep all predictive features because omitting one least important (e.g., precipitation) is expected to lead to poorer model performance.

Figure 1a and Figure 2a both show time-series of NOx and EC. Is this a daily average? If it is, please add the standard deviation for year 2020. This might show that in fact the change is not as apparent as it seems.

Response: Figures 1a and 2a showed the 7-day rolling means of NOx and EC for 2020/2015-2019, respectively. It is not a daily average. Rolling means make the patterns clearer and remove the day-to-day variations generally observed in the ambient environment (Grange et al., 2021).

It is essential that a dependence scatter plot be provided to show the effect of a single feature across the whole dataset. It will also be useful to just take the mean absolute value of the SHAP values for each feature to get a standard bar plot that can be easily interpreted.

Response: The original Figure 4 (now Figure 4a and 4b) shows the distribution of SHAP values. It has demonstrated the relative importance of all the features. The revised Figure 4c and 4d shows the mean absolute value of SHAP.

Reviewer #2

General comments:

In this manuscript, the time series of vehicular emissions of $NO_x$ and EC before and during the 2020 lockdown as well as the averaged time series of $NO_x$ over the same period for the previous four years (i.e., the mean of 2016-2019) were compared and used to train the machine learning model, rebuilding the $NO_x$ and EC in a business-as-usual scenario in 2020. This study improves the understanding of the trend in vehicular emissions and, in particular, in response to the strict Covid-19 lockdown based on the long-term observation dataset and application of advanced mathematical models.

My specific remarks are given below, after a minor revision the manuscript can be accepted.

Response: We thank the Reviewer for the positive comment. Please see the point-to-point response to each comment below.

Specific remarks:

- 1. Line 45: "$NO_2$ concentrations are" --> "$NO_2$ concentration is"

Response: corrected.

- 2. Line 46: "$NO_x$ (NO+$NO_2$) emissions" --> "$NO_x$ (NO+$NO_2$) emission"

Response: corrected.

- 3. Lines 50-51: "elemental carbon (EC) or black carbon is emitted a result of incomplete combustion of fossil fuel (gasoline and diesel) in the internal combustion engine", this description is not sufficiently rigorous. The incomplete combustion of biomass is also the one source of elemental carbon (EC) or black carbon. This description should be rewritten.

Response: Thank you for pointing this out. We have rewritten this to "…Elemental carbon (EC) or black carbon is a major component of fine PM ($PM_{2.5}$) from vehicular emission. EC is emitted as a result of incomplete combustion of gasoline or diesel in the internal combustion engine (Lin et al., 2020; Jia et al., 2021; Lin et al., 2022), with significant health and climate implications (Ramanathan and Carmichael, 2008; Cappa et al., 2012; Rappazzo et al., 2015)…"

- 4. Lines 53-56: The duplicate content "with the recent implementation of high emission standards (e.g., China IV and V)" should be deleted to make sentence more concise.

Response: corrected.

- 5. Line 165: "measures (Grange et al., 2021)Specifically," --> "measures (Grange et al., 2021). Specifically,"

Response: corrected.

- 6. Line 202: "DOY" Please give the full name day of the year (DOY).

Response: corrected.

References:

[revised manuscript text omitted]

---

## Author Response (AR2)

We thank the reviewer for the further comment which helped improve the manuscript. We provide point-to-point repones below in blue, and the revision in the main is in red.

The authors have adequately improved the manuscript. A few minor revisions required before publication, including:
Response: we thank the reviewer for the positive comment.
Line 15: Change "element carbon" to "elemental carbon".
Response: corrected.

Section 3.3: I understand that the authors opted to use RSQ and r to represent model performance. However, there is a need to support this with an error evaluation metric such as RMSE (Root Mean Square error) that measures the average magnitude of the error in the model. Comparing the RMSE of both the MLR and RF models is essential. Please add a section discussing this. Please update the methods section accordingly.
Response: We have now provided RMSE comparison between RF and MLR. We have added more discussion in the revised Sect. 2.2.1.
In the revised Sect. 2.2.1, it now reads, "…the correlation coefficient R and the root mean square error (RMSE) between the time series of measured and predicted pollutants. The performance of Random Forest model was compared to the multilinear regression (MLR) model, in terms of the R value and the RMSE value (Table S1)…." And "…Moreover, the RMSE values are smaller for the Random Forest Model (i.e., 0.27-0.51 (training-testing) $\mu g\ m^{-3}$ and 12.94-29.34 $\mu g\ m^{-3}$ for EC and NOx, respectively) than the MLR (0.96 $\mu g\ m^{-3}$ for EC and 47.6 $\mu g\ m^{-3}$ for $NO_x$; Table S1)…"

**Table S1. Correlation coefficient R and root mean square error (RMSE in $\mu g\ m^{-3}$) between predicted and measured NOx and EC using the random forest model and the multilinear regression model.**

| R (RMSE) | Random Forest | Multilinear regression |
|---|---|---|
| $NO_x$ | 0.89-0.98 (12.94-29.34) | 0.48 (47.6) |
| EC | 0.90-0.98 (0.27-0.51) | 0.45 (0.96) |